# A Novel Lipase from *Lasiodiplodia theobromae* Efficiently Hydrolyses C8-C10 Methyl Esters for the Preparation of Medium-Chain Triglycerides’ Precursors

**DOI:** 10.3390/ijms221910339

**Published:** 2021-09-25

**Authors:** Andre Mong Jie Ng, Renliang Yang, Hongfang Zhang, Bo Xue, Wen Shan Yew, Giang Kien Truc Nguyen

**Affiliations:** 1WIL@NUS Corporate Laboratory, Wilmar International Limited, Centre for Translational Medicine, 14 Medical Drive, Singapore 117599, Singapore; andre.ng@u.nus.edu (A.M.J.N.); renliang.yang@sg.wilmar-intl.com (R.Y.); hongfang.zhang@sg.wilmar-intl.com (H.Z.); 2Wilmar Innovation Centre, Wilmar International Limited, 28 Biopolis Road, Singapore 138568, Singapore; 3Department of Biochemistry, Yong Loo Lin School of Medicine, National University of Singapore, 8 Medical Drive, Singapore 117596, Singapore; xuebo@nus.edu.sg; 4NUS Synthetic Biology for Clinical and Technological Innovation, Centre for Life Sciences, National University of Singapore, 28 Medical Drive, Singapore 117456, Singapore; 5NUS Synthetic Biology Translational Research Programme, Centre for Translational Medicine, National University of Singapore, 14 Medical Drive, Singapore 117599, Singapore

**Keywords:** enzyme discovery, lipase, medium-chain triglycerides, methyl esters, sustainability, zymogram

## Abstract

Medium-chain triglycerides (MCTs) are an emerging choice to treat neurodegenerative disorders such as Alzheimer’s disease. They are triesters of glycerol and three medium-chain fatty acids, such as capric (C8) and caprylic (C10) acids. The availability of C8–C10 methyl esters (C8–C10 ME) from vegetable oil processes has presented an opportunity to use methyl esters as raw materials for the synthesis of MCTs. However, there are few reports on enzymes that can efficiently hydrolyse C8–C10 ME to industrial specifications. Here, we report the discovery and identification of a novel lipase from *Lasiodiplodia theobromae* fungus (LTL1), which hydrolyses C8–C10 ME efficiently. LTL1 can perform hydrolysis over pH ranges from 3.0 to 9.0 and maintain thermotolerance up to 70 °C. It has high selectivity for monoesters over triesters and displays higher activity over commercially available lipases for C8–C10 ME to achieve 96.17% hydrolysis within 31 h. Structural analysis by protein X-ray crystallography revealed LTL1’s well-conserved lipase core domain, together with a partially resolved N-terminal subdomain and an inserted loop, which may suggest its hydrolytic preference for monoesters. In conclusion, our results suggest that LTL1 provides a tractable route towards to production of C8–C10 fatty acids from methyl esters for the synthesis of MCTs.

## 1. Introduction

Caprylic (C8) and capric (C10) triacylglycerols, commonly known as medium-chain triglycerides (MCTs), are widely used in food ingredients, nutritional supplements, and medical treatment for neurological diseases. A major therapeutic option is the use of MCTs in the treatment of Alzheimer’s disease (AD). AD is an age-associated and neurodegenerative condition that progressively reduces the individual’s cognitive memory and language abilities. Besides being a major health problem, AD has severe physical, financial, psychological, and social consequences for the patient and the patient’s family. Due to the ketogenic nature of MCTs (increasing production of ketone bodies in the liver, leading to the availability of ketonic fuel for neurons), the utility of MCTs in the treatment of AD is currently being actively investigated [1]. In several studies, MCTs were found to improve memory and visual cognition [2,3] and slow down neurodegeneration [4,5]. Besides treating AD, MCTs have also been implicated in reducing the occurrence of body hyperlipidaemias, inflammation, obesity, and oxidative stress [6,7,8,9], amongst other metabolic diseases.

Industrial manufacturing of C8 and C10 MCTs largely involves the direct esterification of glycerol and medium-chain fatty acids under high pressure and temperature to form tricaprylin and tricaprin [10]. Feedstocks of these MCTs largely rely on C8 and C10 fatty acids and come from two main sources: (1) fractional distillation of hydrolysed vegetable oil at high temperature and pressure, and (2) hydrolysis of C8-C10 methyl esters (C8-C10 ME), which are generally available as a mixture containing both C8 and C10 methyl esters. Materials sourced from the latter are widely preferred due to their low cost and availability in large quantities via transesterification of palm kernel oil and coconut oil [11,12].

To obtain a steady source of C8 and C10 fatty acids, hydrolysis of C8-C10 ME can be carried out enzymatically with lipases and esterases. These enzymes catalyse the hydrolysis of ester bonds of tri-, di-, and monoglycerides into fatty acids and alcohol [13]. However, commercially available lipolytic enzymes from microbial strains *Candida antarctica* [14,15], *Rhizomucor miehei* [16], and *Thermomyces lanuginosus* [17,18] are not substrate-specific and have not been shown to efficiently hydrolyse C8-C10 ME. Recently, an improved zymographic method coupled with liquid chromatography-mass spectrometry (LC-MS) [19] had demonstrated that the detection of lipase and esterase activities could be improved by up to 1000-fold. The improved method can be utilised to identify novel pools of lipolytic enzymes, which might have otherwise gone undetected, and facilitate our search for lipases specific to C8-C10 ME hydrolysis.

Here in this study, we present a novel lipase originally isolated from the filamentous fungal strain *Lasiodiplodia theobromae* (*L. theobromae*) from our Wilmar International in-house culture collection. We first screened several microbial species for their growth on minimal media with tri- and mono-glycerides as a sole carbon and energy source. *L. theobromae* lipase 1 (LTL1) displayed the highest activity during zymographic analysis on chromogenic agar supplemented with tri- and mono-glycerides. Fed-batch fermentation was conducted to recombinantly express LTL1 in *Pichia pastoris* (*P. pastoris*) to characterise LTL1’s substrate, pH, and temperature specificities. Upon comparison, LTL1 outperformed various commercially available lipolytic enzymes and was able to match up against industrially relevant hydrolytic benchmarks for C8-C10 ME. Lastly, protein X-ray crystallographic studies were carried out to understand LTL1’s hydrolytic preference for monoesters. Taken together, LTL1 is a very promising candidate enzyme that can be used in the hydrolysis of medium-chain length monoesters for the industrial synthesis of MCTs.

## 2. Results and Discussion

### 2.1. Identification of LTL1 from L. theobromae with Zymography-LC-MS

To identify novel lipolytic enzymes capable of efficient C8-C10 ME hydrolysis, we performed a candidate screen with rhodamine B within our in-house culture collection and selected the environmental fungus *L. theobromae*. This fungal strain was able to utilise olive oil as its sole carbon and energy source for growth (Figure 1A). After culturing for five days in liquid minimal media, the supernatant of *L. theobromae* culture was separated with native-polyacrylamide gel electrophoresis (Native-PAGE) followed by zymographic analysis on chromogenic phenol red agar plate supplemented with monolaurin or tributyrin, as previously described [19]. Tributyrin was used as the primary screening substrate, while monolaurin was used as an additional substrate to identify lipolytic enzymes with a preference for medium-chain carbon lengths. Hydrolysis of tributyrin or monolaurin releases free fatty acids into the agar, causing a decrease in pH and colouration change in the phenol red pH indicator from red (neutral) to yellow (acidic). From our experiments, a total of four lipolytic bands displaying putative lipolytic activities were detected with our zymogram detection method (Figure 1B). These bands were excised, tryptic digested, and analysed with LC-MS. Sequenced peptides were referenced back to the available translated genomic sequence of *L. theobromae* (NCBI Genome ID 54334) [20] to identify open reading frames (ORFs). Further analyses resulted in the identification of a novel lipase from *L. theobromae*; for brevity and clarity, this lipase will be referred to as LTL1 for the remainder of the study (Figure 1C). The remaining three putative lipolytic enzymes seen in Figure 1B were evaluated to show low activity for C8-C10 ME hydrolysis and were not pursued further in this study (data not shown). LTL1, we believe, was incorrectly annotated as an FMN binding protein in the available databases. LTL1 also shares 73% sequence identity with an uncharacterised functionally unassigned Lipase B from *Diplodia seriata* (Figure 1D). 

### 2.2. Expression of Lipolytic Enzyme in Fed-Batch Fermentation

Heterologous protein expression was carried out for the enzyme for functional characterisation: the gene encoding *P. pastoris* codon-optimised LTL1 was cloned into a pPIC9K vector, under the control of methanol inducible *AOX1* promoter, for extracellular expression in *P. pastoris*. Following the manufacturer’s protocol [21], a glycerol-batch followed by glycerol and methanol fed-batch fermentation phases were conducted with a Minifors 2 bench-top Bioreactor. As shown in Figure 2A, the wet cell mass (WCM) reached approximately 220 g/kg at the end of the glycerol fed-batch phase, followed by methanol fed-batch for approximately ~120 h. The WCM and protein concentration of the LTL1 recombinant expression strain reached approximately 500 g/kg and 1.74 g/L at the end of the fermentation, respectively. Figure 2B shows that the crude supernatant has more than 90% purity of LTL1 that can be used directly for functional characterisation and protein X-ray crystallisation without further purification.

### 2.3. Characterising LTL1 Substrate, pH, and Temperature Specificities

To determine the substrate specificity of LTL1, we conducted hydrolysis assays against a panel of substrates including triesters (coconut oil, olive oil, and tributyrin) and monoesters (monobutyrin, monoolein, and vinyl laurate). Among the group of triesters studied, lower hydrolytic activities were detected for olive oil and coconut oil, while higher hydrolytic activity was detected for tributyrin, which indicates the stronger preference for shorter-chain triesters. In contrast, the hydrolytic activities of LTL1 were much higher for monoesters compared to triesters (Figure 3A). Consistent with its preference for shorter-chain substrates, LTL1 displayed higher activity for monobutyrin as compared to monoolein. Within expectations, the enzyme also displayed the highest activity for vinyl laurate compared to all the other substrates, due to the irreversible hydrolysis of the labile vinyl alcohol to acetaldehyde.

To determine the optimal pH and temperature profiles of LTL1, hydrolysis assays with tributyrin were conducted over a range of pH (from 3.0 to 10.0) and temperatures (from 20 to 90 °C). LTL1 hydrolytic activity was maintained throughout the pH range of 3.0 to 9.0 (Figure 3B), with peak hydrolysis activity at 70 °C (Figure 3C). Its high thermal tolerance indicates that LTL1 can be useful for the hydrolysis of substrates with high melting points; the corollary expectation can be drawn for LTL1-mediated hydrolysis to be conducted at higher temperatures to reduce reaction times as well as other industrially aligned production parameters.

### 2.4. LTL1 Efficiently Hydrolyses C8-C10 ME

In an attempt to establish the industrial relevance of LTL1, we compared the catalytic activities of the enzyme vis-à-vis hydrolysis of vinyl laurate and C8-C10 ME against commercially available lipolytic enzymes, including *Thermomyces lanuginosus* lipase (TLL), *Rhizomucor miehei* lipase (RML), *Candida antarctica* lipase B (CALB), and Eversa 2.0. The C8-C10 ME used in this study was a mixture of 53% C8 ME and 46% C10 ME, respectively (provided by Wilmar International Limited). These hydrolytic reactions were carried out at 37 °C, pH 7.2. We observed that LTL1 outperformed all commercial enzymes used in this study in the hydrolysis of vinyl laurate and C8-C10 ME, displaying activities of 11,000 and 3000 U/mg, respectively (Figure 4). The specific activity (U/mg) of LTL1 was 3 to 10-fold higher than those of the commercial lipases used in this study.

For industrial applications, a greater than 95% hydrolysis of C8-C10 ME to their corresponding acids is required to avoid the additional process of distillation (which would result in increased production costs). The degree of hydrolysis is usually monitored by measuring the acid value (AV) of the hydrolysed product. In this context, a completely hydrolysed (100%) C8-C10 ME mixture would give an AV of 365. To determine if LTL1 can achieve the minimal 95% hydrolysis of C8-C10 ME, we performed the hydrolysis reactions and compared the performance of LTL1 against two commercial lipases, Eversa 2.0 and RML, respectively. Over two days, 100 parts per million (ppm) dosage (*w*/*w* based on oil weight) of LTL1 was able to perform 96% of C8-C10 ME hydrolysis within 31 h, achieving an AV of 353 (Table 1 and Figure 5). In the same timeframe, Eversa 2.0 and RML achieved AV of 241 and 271, respectively. Despite the use of more Eversa 2.0 enzyme (an increased Eversa 2.0 dosage of 750 ppm), hydrolysis of C8-C10 ME could only reach 329 AV (90% hydrolysis). These results demonstrated that LTL1 is a highly efficient lipase for C8-C10 ME hydrolysis, as compared to commercially available lipases (such as Eversa 2.0 and RML); our study also suggested that LTL1 is a suitable candidate for industrial applications such as the industrial synthesis of MCTs. 

### 2.5. Protein X-ray Crystallography of LTL1

To gain further insights into its selectivity for monoesters over triesters, we solved the 3D structure of LTL1 by protein X-ray crystallography (Table 2). The crystal structure of LTL1 consists of two nearly identical polypeptide chains covering amino acid residues 83-449 (chain A) and 84-449 (chain B) of *L. theobromae* lipase B (UniprotKB: A0A5N5DNA6), respectively. These residues fold into lipase core domains (Figure 6A, yellow and cyan) with N-terminal extensions (orange and pink) and inserted loops (magenta and blue) and are in close contact with each other through an extensive interfacial area of 1229 Å^2^ according to PISA analysis [22]. For simplicity, future discussions will be focused on chain A, unless otherwise stated. The N-terminal portion of the expressed LTL1 (residues 19–82) could not be resolved in the structure, likely due to proteolysis during crystallisation. A model of non-truncated LTL1 was generated by AlphaFold2 [23] through the online Google Colaboratory (ColabFold: AlphaFold2 w/MMseqs2) [24]. The model predicts that LTL1 has an N-terminal subdomain residing above the active site, which seems to be plausible given its accurate reconstruction of the core domain and correct prediction of the secondary structure (i.e., the two alpha helices, coloured orange in the X-ray structure and dark green in the model) of the N-terminal extension observed in the crystal structure (Figure 6B). 

A Dali search [25,26] with the LTL1 structure reveals that closest structural homologs of LTL1 are lipases B from *Aspergillus fumigatus* (AFLB, PDB ID: 6IDY) [27] and *C. antarctica* (CALB, representative PDB ID: 5A71) [28], with reported RMSDs of 2.08 and 2.16 Å, respectively, by the mTM-align server [29]. As shown in Figure 6C,D, these lipases possess well-conserved core domains. Prominent structural differences among them lie in their N-termini and inserted loops, or the lack of. When compared with CALB, both LTL1 and AFLB have an extra N-terminal subdomain, albeit with a different size and conformation, and an inserted loop flanked by two cysteines that form a disulphide bond. The loop in AFLB flips inwards, covering the active site, whereas the loop in LTL1 bends outwards, leaving LTL1 in an “open state” that has similarly been observed in CALB. With the help of the intact N-terminal subdomain, LTL1’s loop could potentially reorientate and at least partially cover the active site, as predicted by the AlphaFold2 model (Figure 6B). The presence of the additional N-terminal subdomain and inserted loop covering the active site seems to suggest that steric effects could be part of the reason for LTL1’s preference for slimmer monoesters over bulkier triesters. In addition, these structural elements may also play an important role in regulating LTL1’s hydrolytic activities, as likewise demonstrated by their counterparts in AFLB [27].

## 3. Materials and Methods

Reagents described in all experiments were sourced from Sigma-Aldrich (Singapore) unless stated otherwise. C8-C10 ME was obtained from Wilmar Oleochemicals Limited (Lianyungang, China). Monoolein was obtained from Kao Chemicals, Japan. CALB (Lipozyme, Sigma-Aldrich L3170), Eversa Transform 2.0 (Sigma-Aldrich SAE0065), and RML (Palatase 20,000 L, Sigma-Aldrich L4277), TLL (Lipolase, Sigma-Aldrich L0777) were used as lipolytic enzyme standards.

### 3.1. Strains and Culture Conditions

*L. theobromae* screened for lipolytic enzymes was obtained from an in-house culture collection. They were kept in glycerol stocks at −80 °C and maintained on YPD (10 g/L yeast extract, 20 g/L peptone, 20 g/L dextrose) agar (20 g/L) at 4 °C. When needed, *L. theobromae* was cultured in YPD liquid at 220 revolutions per minute (rpm), 30 °C (Yihder Orbital Shaking Incubator-Vertical type LM-80D~LM-575, Xinbei, Jiangsu, China). 

*Escherichia coli* DH5α used for vector propagation was obtained from Thermo Fisher, Singapore. They were kept in glycerol stocks at −80 °C and maintained on Lysogeny Broth (LB) agar (10 g/L tryptone, 5 g/L yeast extract 50 g/L sodium chloride) agar (15 g/L) at 4 °C. When needed, DH5α was cultured in LB liquid at 220 rpm, 37 °C.

*P. pastoris* GS115 used for protein expression was obtained from Invitrogen, Singapore. They were kept in glycerol stocks at −80 °C and maintained on YPD agar 4 °C. When needed, GS115 was cultured in YPD liquid at 220 rpm, 30 °C.

Liquid culture used to screen for lipolytic-producing enzyme contained minimal media (0.73 g/L disodium phosphate, 0.35 g/L potassium dihydrogen phosphate, 0.1 g/L magnesium sulphate heptahydrate, 0.75 g/L ammonium nitrate, 0.25 g/L sodium bicarbonate, 0.002 g/L manganese sulphate and 0.02 g/L iron (II) sulphate) [19], where appropriate, supplemented with 10 g/L olive oil, 50 g/L Lysogeny Broth or 10 mg/L Rhodamine B and cultured at 28 °C, 120 rpm, for 48 to 144 h.

### 3.2. Detection and Identification of LTL1 with Zymography-LC-MS

Detection of lipase activity was performed with Zymography-LC-MS, as previously described [19]. Briefly, the pool of lipolytic enzymes produced in the culture media by the *L. theobromae* was separated in native conditions with Native-PAGE [30]. Native-PAGE gels with separated lipolytic enzymes were rinsed with distilled water and equilibrated with 0.5 mM Tris-HCl (pH 8.0) at 4 °C. Equilibrated gels were developed on chromogenic phenol red agar plates supplemented with tributyrin or monoolein at 37 °C for 5 to 60 min. Lipolytic zones which displayed colour change were excised and in-gel digested into peptides [19,31], where these pieces were de-stained, reduced, alkylated, and tryptic digested.

Digested peptides were resuspended in acetonitrile and analysed with an Ultimate 3000 High-Performance Liquid Chromatography system (Thermo Scientific, Waltham, MA, USA) coupled to Q Exactive Plus Orbitrap Mass Spectrometer (Thermo Scientific, Waltham, MA, USA), as previously described [32]. Then, 5 µL of digested protein samples were separated on the Acclaim PepMap RSLC C18 Column (300 µm × 150 mm, 3 µm, 100 Å) (Thermo Scientific, Waltham, MA, USA). The flow rate was set at 10 µL/min, with 0.1% *v*/*v* formic acid (Merck, Kenilworth, NJ, USA) in distilled water as solvent A and 0.1% *v*/*v* formic acid in acetonitrile (Merck, Kenilworth, NJ, USA) as solvent B. The running gradient was as follows: 5% solvent B from 0 to 2 min, 5 to 70% solvent B from 2 to 21 min, 70 to 99% solvent B from 21 to 21.3 min, 99% solvent B from 21.3 to 25.5 min, 99 to 1% solvent B from 25.5 to 26 min, and 1% solvent B from 26 to 30 min. Data acquired on the mass spectrometer were operated in full scan mode followed by data-dependent tandem mass spectrometry (MS/MS) acquisition over a mass range of 300–1800 *m*/*z*.

For peptide identification on the LC-MS, data analysis was performed with PEAKS Studio 7.5 (build 20150615) as previously described [33]. Sequenced peptides were identified, and de novo assembled. Assembled peptides were searched against the UniProt reference proteome database (559,213 sequences, updated on 12 September 2019) with a mass error tolerance for parent ion mass as ±10 ppm and with fragment ion as ±0.5 Da. Methionine oxidation and protein N-terminal acetylation were chosen as variable modifications, and cysteine alkylation by iodoacetamide was chosen as a fixed modification. The protease was specified as trypsin with three maximum missing cleavages.

Images were visualised by open-source software Fiji (ImageJ) [34]. The brightness tool was used in auto-scaled mode to optimise the visualisation of images (Image > Adjust > Brightness/Contrast > Auto).

### 3.3. Strain Construction and Transformation into Expression Strain

*P. pastoris* GS115-pPIC9K-*LTL1* was constructed by inserting a full-length codon optimised sequence of *LTL1* for *P. pastoris* into the open reading frame of α-factor secretion signal sequence of expression vector pPIC9K (Invitrogen, Waltham, MA, USA) via the restriction sites *SnaBI* and *EcoRI*. Vector propagation was carried out in DH5α and selected with LB agar supplemented with kanamycin (50 µg/mL). Transformed colonies were screened for the correct inserts with colony polymerase chain reaction (PCR) and subsequently verified with Sanger Sequencing. Verified pPIC9K-*LTL1* vector were linearised at *SalI* and transformed into *P. pastoris* GS115 competent cells by electroporation [35]. Transformed GS115 were selected on RDB agar (1 M sorbitol, 20 g/L glucose, 3.4 g/L of yeast nitrogen base, 10 g/L ammonium sulphate, 0.4 mg/L biotin, 20 g/L agar and 0.05 g/L of each L-glutamic acid, L-methionine, L-lysine, L-leucine, and L-isoleucine) plates, verified for *LTL1* sequence with genomic DNA PCR, and subsequently maintained on BMGY (0.1 M potassium phosphate buffer, pH 6.0, 34 g/L yeast nitrogen base without ammonium sulphate and amino acid, 10 g/L yeast extract, 20 g/L peptone, 100 g/L ammonium sulphate, 0.4 mg/L biotin, 1% *v*/*v* glycerol) agar (20 g/L) plates.

### 3.4. Fed-batch Fermentation Conditions

The seed culture of GS115-pPIC9K-*LTL1* was prepared in BMGY medium. Fed-batch fermentation was conducted in 1 L bioreactor (INFORS HT Minifors 2, Bottmingen, Basel, Switzerland) containing 400 mL of basal salt medium (26.7 mL/L 85% *v*/*v* phosphoric acid, 0.93 g/L calcium sulphate, 18.2 g/L potassium sulphate, 14.9 g/L magnesium sulphate heptahydrate, 4.13 g/L potassium hydroxide, 40 g/L glycerol) supplemented with 4.35 mL PTM (6 g/L copper (II) sulphate, 80 mg/L sodium iodide, 3 g/L manganese sulphate monohydrate, 200 mg/L sodium molybdate dihydrate, 20 mg/L boric acid, 500 mg/L cobalt chloride, 20 g/L zinc chloride, 65 g/L iron (II) sulphate heptahydrate, 200 mg/L biotin, and 5 mL/L sulphuric acid), as per the manufacturer’s protocol [21]. The pH was maintained at 6.0 and adjusted with 28% *v*/*v* ammonium hydroxide. The airflow rate was set at 1 vvm (vessel volume per minute), the agitation speed was set at 1000 rpm, and the dissolved oxygen (DO) was maintained at greater than 30% saturation through a mixed supply of compressed air (Ekom DK40 2V, Singapore) and pure oxygen (Air Liquide, Singapore). The entire fermentation was conducted in three phases. Phase I was the glycerol batch phase, where vessel temperature was maintained at 30 °C until glycerol was fully consumed, indicated by a sharp spike of DO. Phase II was the glycerol fed-batch, 50% *v*/*v* glycerol with 12 mL/L of PTM was fed for an additional 4 h. Phase III was the methanol fed-batch, vessel temperature was lowered to 28 °C and pure methanol (Fisher Scientific, Singapore) with 12 mL/L of PTM was fed continuously to induce protein production for 96 h. The entire fermentation was periodically monitored for biomass (WCM), lipolytic activity, and protein concentration.

### 3.5. Enzyme Activity Characterisation Conditions

Lipolytic enzymatic activity was measured using the appropriate tri- and mono-esters as substrate. One unit of activity was defined as the amount of enzyme catalysing the lipolysis of 1 µmol fatty acid per min [36].

For pH specificity, hydrolysis was performed using 20 µL of LTL1, 120 µL distilled water, 60 µL of different 0.2 M buffer ranging from pH 3.0 to 10.0 and 100 µL of tributyrin in a 2 mL tube. The reaction was performed at 37 °C, 2000 rpm (Eppendorf ThermoMixer, Hamburg, Germany) for 15 min. The reactions were stopped by adding an equal volume of pure ethanol, and free fatty acid liberated was neutralised with 50 mM sodium hydroxide.

For temperature specificity, hydrolysis was performed using 20 µL of LTL1, 120 µL distilled water, 60 µL 0.2 M Tris-HCl buffer (pH 7.2), and 100 µL of tributyrin in 2 mL tubes. The reaction was performed at temperatures ranging from 20 to 100 °C, 2000 rpm for 15 min. The reactions were stopped by adding an equal volume of pure ethanol and free fatty acid liberated was neutralised with 50 mM sodium hydroxide.

For substrate specificity, hydrolysis was performed using 20 µL of LTL1, 120 µL distilled water, 60 µL 0.2 M Tris-HCl buffer (pH 7.2) and 100 µL of different substrates (coconut oil, vinyl laurate, olive oil, monoolein, tributyrin, or monobutyrin) in a 2 mL tube. The reaction was performed at 37 °C, 2000 rpm for 15 min. The reactions were stopped by adding an equal volume of pure ethanol, and free fatty acid liberated was neutralised with 50 mM sodium hydroxide.

For comparison with commercial lipases, hydrolysis of vinyl laurate and C8-C10 ME were carried out with TLL, RML, CALB, and Eversa. The reaction was performed using 20 µL of enzyme, 120 µL water, 60 µL 0.2 M Tris-HCl buffer (pH 7.2), and 100 µL substrates in a 2 mL tube. The reaction was performed at 37 °C, 2000 rpm for 15 min. The reactions were stopped by adding an equal volume of pure ethanol, and free fatty acid liberated was neutralised with 50 mM sodium hydroxide.

To screen LTL1 with industrial hydrolysis conditions, C8-C10 ME hydrolysis reactions were performed with 20 g methyl esters, 40 g distilled deionised water, and a corresponding dosage of lipases at 45 °C, 50 mbar of vacuum. After hydrolysis, the mixture was centrifuged at 7000 *rcf* for 30 min. The acid value of the top organic layer was determined with the AOCS Cd 3d-63 method [36].

### 3.6. Enzyme Preparation for Protein X-ray Crystallisation

The *P. pastoris* fermentation culture was centrifuged at 8000 *rcf*, 4 °C for 30 min and the supernatant was harvested, and the cell pellet was discarded. The supernatant was then filtered through a 0.22 μm PES (polyethersulfone) membrane (Sartorius, Germany) to remove any remnant microbial impurities. The supernatant was concentrated with Amicon Ultra-15 10 kDa Centrifugal Filter Unit (Millipore, Burlington, MA, USA) and buffer was exchanged to 10 mM Tris-HCl, pH 8.0 with 150 mM sodium chloride. The final concentration of LTL1 from fermentation was calculated to be approximately 17.48 g/L.

### 3.7. Protein X-ray Crystallisation Conditions

Crystallisation of LTL1 was conducted using an NT8 drop setter for protein crystallisation (Formulatrix, New Bedford, MA, USA) and 96-well 2-drop MRC sitting-drop vapor diffusion plates (Swissci, Neuheim, Zug, Switzerland). Next, 100 nL of lipase solution (17.48 g/L) was mixed with 100 nL of a precipitant (60% *v*/*v* 2-methyl-2,4-pentanediol, 100 mM sodium acetate, pH 4.6, 10 mM calcium chloride) and incubated at 20 °C. Crystals were cryoprotected with a 1:1 mixture of the same precipitant and saturated sodium malonate before being flash-frozen in liquid nitrogen. Diffraction data were collected at the MX1 beamlines of the Australian Synchrotron, part of ANSTO [37], and were processed using XDS [38]. To solve the structure by molecular replacement, chain B of PDB 6IDY was deployed as the search model in PHASER [39], and the resultant solution was refined in REFMAC [40] and Phenix [41], followed by manual building in Coot [42]. MolProbity [43] was employed for model quality check. PyMOL (The PyMOL Molecular Graphics System, Version 2.4.1 Schrödinger, LLC) was used for structure visualisation and figure generation.

## 4. Conclusions

In conclusion, while screening several lipase-producing microbial species, LTL1 displayed the highest activity during zymographic analysis on chromogenic agar supplemented with tributyrin and monoolein. While characterising enzyme pH and temperature profiles, LTL1’s hydrolytic activities were detectable within the pH range of 3.0 to 9.0 and were thermotolerant of up to 70 °C. As for substrate specificity, LTL1 prefers the hydrolysis of monoesters and monoglycerides over triglycerides. Upon comparison with various commercially available lipolytic enzymes, LTL1 outperformed and matched up to industrially relevant hydrolytic benchmarks for C8-C10 ME hydrolysis (greater than 95% hydrolysis). Lastly, the protein X-ray crystallographic structure of LTL1 revealed the presence of the additional N-terminal subdomain and inserted loop covering the active site, which seems to suggest that steric effects could be part of the reason for LTL1’s preference for slimmer monoesters over bulkier triesters. Taken together, our results suggest that LTL1 provides a tractable route towards the production of C8–C10 fatty acids from methyl esters for the synthesis of MCTs. We believe that our newly discovered enzyme could be broadly applied as a greener choice to prepare the precursors for MCTs synthesis for therapeutic applications.

## 5. Patents

A patent for the use of LTL1 for C8-C10 ME hydrolysis was filed in Singapore at the Intellectual Property Office of Singapore (application number 10201914033Y) on 31 December 2019 and published in the United States of America (publication number US2021198704A1 [44]) on 1 July 2021 and in India (application number 202014055387A) on 2 July 2021.

## Figures and Tables

**Figure 1 ijms-22-10339-f001:**
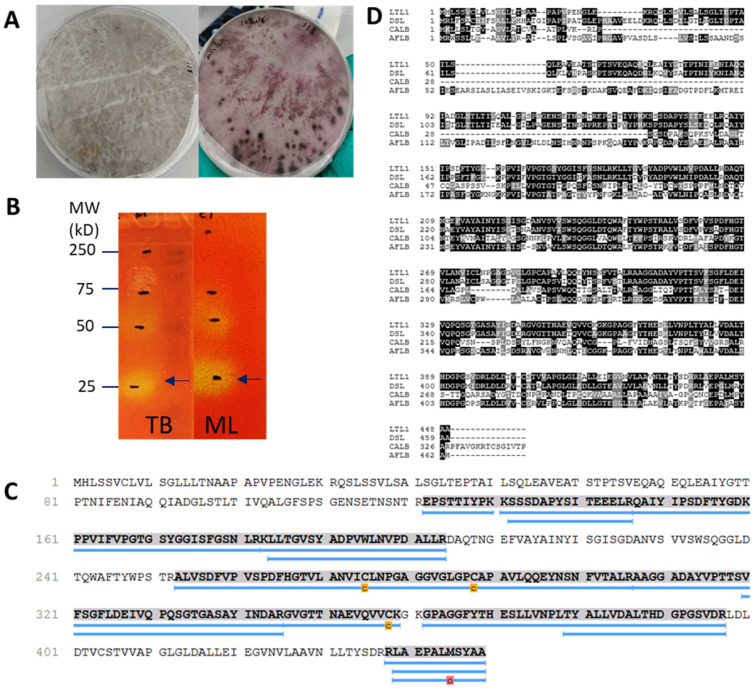
Identification of LTL1 through Zymography-LC-MS. (**A**) Screening of *L. theobromae* on minimal agar supplemented with 1% *v*/*v* olive oil with and 10 µg/mL rhodamine B; (**B**) Zymograms of concentrated *L. theobromae* culture supernatant developed on chromogenic agar supplemented tributyrin (TB) and monolaurin (ML). Lipolytic activity is indicated by a colouration change in the phenol red pH indicator from red or orange to yellow. Arrow indicates LTL1 excised band from zymography and sequenced with LC-MS. (**C**) Peptide mass fingerprinting alignment with LC-MS achieved 53% coverage from 14 unique LTL1 peptides (post-translational modifications indicated by carbamidomethylation (c) and oxidation (o)) sequenced with LC-MS and aligned to *L. theobromae*’s translated genome. (**D**) Multiple gene alignment of LTL1 with lipases identified in *Diplodia seriata* (DSL), *Candida antarctica* (CALB) and *Aspergillus fumigatus* (AFLB).

**Figure 2 ijms-22-10339-f002:**
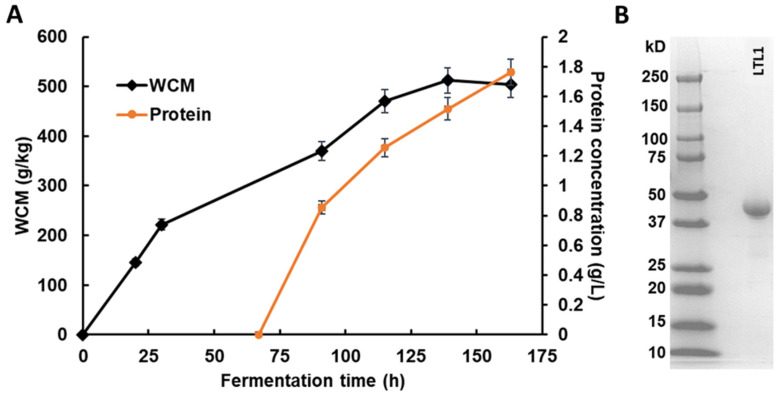
Heterologous protein expression of LTL1 in *P. pastoris* with fed-batch fermentation. (**A**) Cell growth and protein concentration for LTL1 fed-batch fermentation and (**B**) LTL1 expressed by fed-batch fermentation separated by SDS-PAGE. The wet cell mass and protein concentration of the LTL1 recombinant expression strain reached approximately 500 g/kg and 1.74 g/L, respectively. Fermentation culture separated on SDS-PAGE separated approximately 90% purity of LTL1, which was used for protein X-ray crystallisation without further purification.

**Figure 3 ijms-22-10339-f003:**
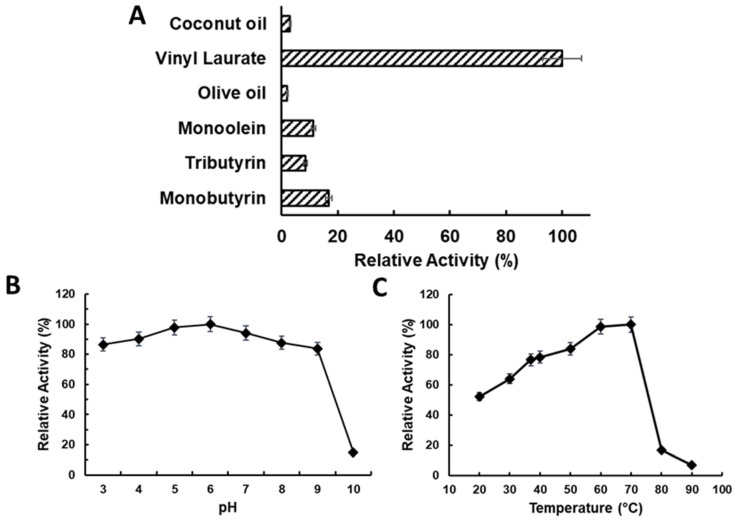
Relative activity of LTL1 hydrolysis assays for (**A**) substrate; (**B**) pH, and (**C**) temperature specificity. LTL1 displayed the highest hydrolysis preference for vinyl laurate among other substrates (coconut oil, olive oil, monoolein, tributyrin, and monobutyrin), while maintaining stable hydrolysis of tributyrin between from pH 3.0 to 9.0, and was thermostable up to 70 °C.

**Figure 4 ijms-22-10339-f004:**
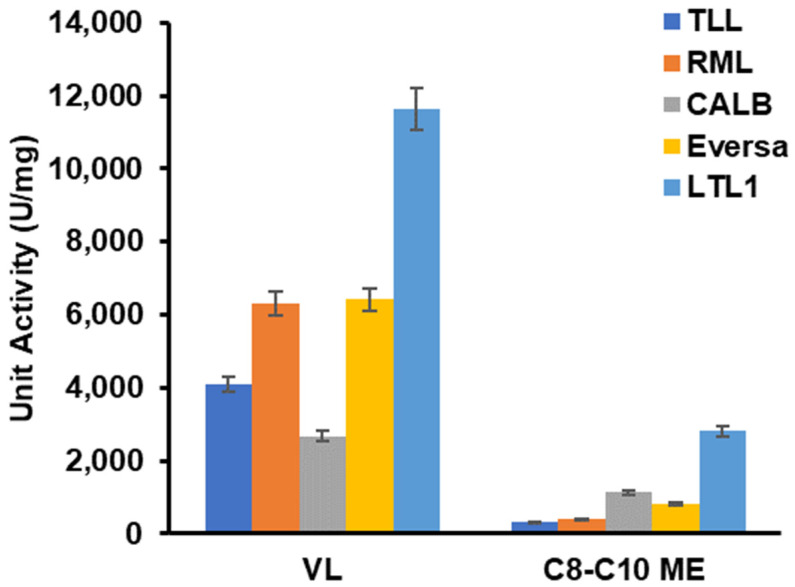
Hydrolysis of vinyl laurate (VL) and C8-C10 ME with LTL1 and commercially available lipases (TLL, RML, CALB, and Eversa 2.0). Over 15 min of reaction time, LTL1 outperformed other commercial enzymes with a normalised unit activity (U/mg) of about 11,000 and 3000 for vinyl laurate and C8-C10 ME, respectively.

**Figure 5 ijms-22-10339-f005:**
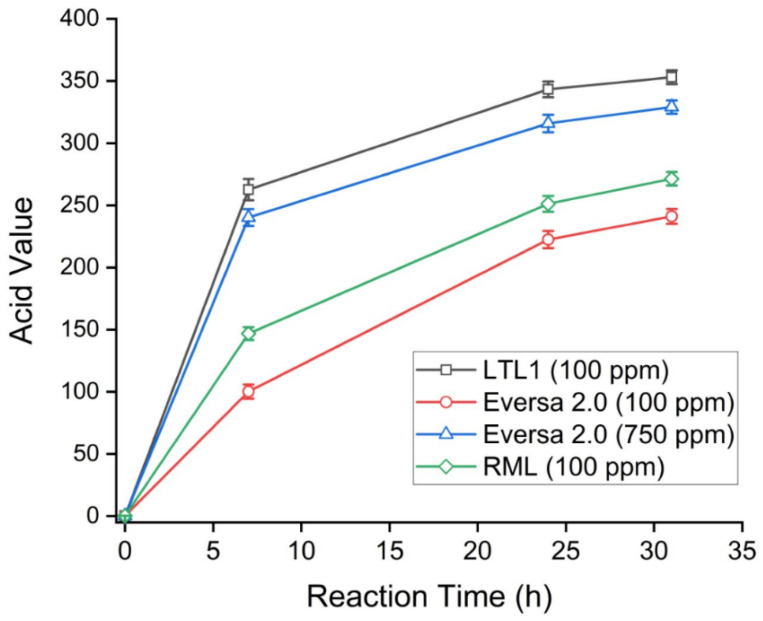
Time-course hydrolysis of C8-C10 ME by LTL1, RML, and Eversa 2.0. Enzyme dosage is 100 or 750 ppm (*w*/*w* based on oil weight). The reaction was conducted as follows: 20 g of C8-C10 ME, 40 mL of water, 100 to 750 ppm of lipases, 45 °C, under 55 mbar vacuum.

**Figure 6 ijms-22-10339-f006:**
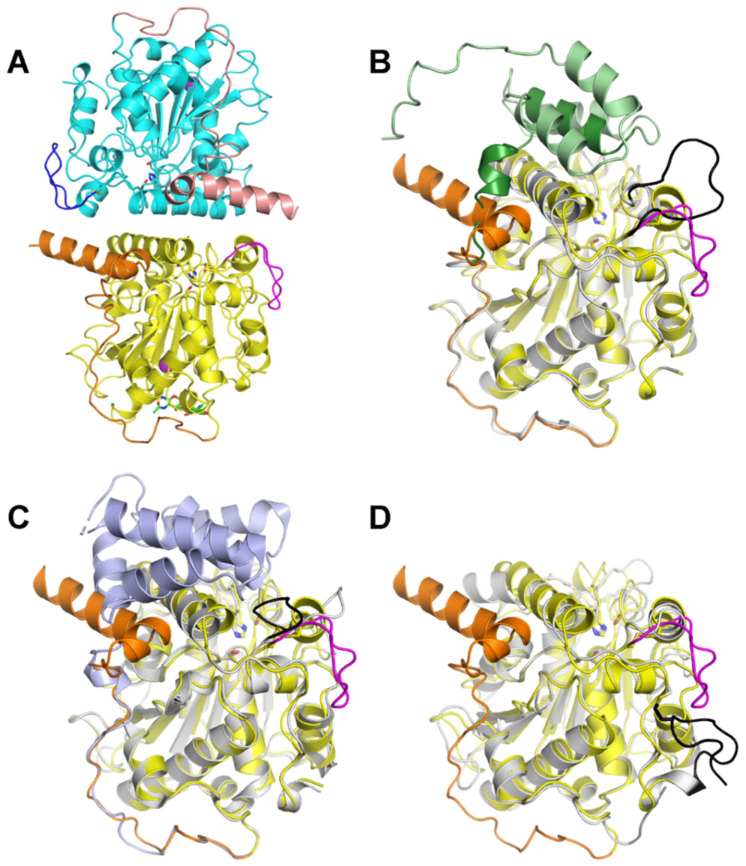
Structures of LTL1 and its closest homologs. (**A**) X-ray structure of LTL1. The two polypeptide chains in the asymmetric unit are coloured differently for chain A (core: yellow, N-terminal extension: orange; inserted loop: magenta) and chain B (core: cyan, N-terminal extension: pink, inserted loop: blue). Sticks are shown to highlight catalytic triads (same colour as the cores) and glycans (green). (**B**) Comparison between LTL1’s X-ray structure (chain A, presented as in A) and an AlphaFold2-predicted model (core: grey, N-terminal subdomain: green, inserted loop: black) that includes both the resolved (83–109, dark green) and unresolved residues (19–82, light green) of LTL1. (**C**) Superimposed structures of LTL1 (chain A, presented as in A) and AFLB (PDB ID: 6IDY, core: grey, N-terminal subdomain: slate, inserted loop: black). (**D**) Superimposed structures of LTL1 (chain A, presented as in A) and CALB (PDB ID: 5A71, core: grey, C-terminal extension: black).

**Table 1 ijms-22-10339-t001:** Hydrolysis of C8-C10 ME by LTL1, RML and Eversa 2.0.

Time (h)	Acid Value (Max Value = 365)			
	LTL1 (100 ppm)	Eversa 2.0 (100 ppm)	Eversa 2.0 (750 ppm)	RML (100 ppm)
0	0.26 (±0.04)	0.26 (±0.04)	0.26 (±0.04)	0.26 (±0.04)
7	262.62 (±8.56)	100.15 (±5.66)	240.25 (±6.73)	146.84 (±5.21)
24	343.26 (±6.32)	222.52 (±6.89)	315.83 (±7.02)	251.21 (±6.39)
31	353.02 (±5.53)	241.16 (±5.95)	329.01 (±5.36)	271.34 (±5.47) W

**Table 2 ijms-22-10339-t002:** X-ray crystallographic data collection, refinement, and validation statistics.

PDB ID	7V6D
**Data collection**	
Wavelength	0.95373
Resolution range	29.49-2.50 (2.59-2.50)
Space group	I 2 2 2
Unit cell	65.43 162.64 166.67 90 90 90
Total reflections	426,375 (42,857)
Unique reflections	31,232 (3060)
Multiplicity	13.7 (14.0)
Completeness (%)	99.88 (100.00)
Mean I/sigma(I)	12.04 (2.20)
Wilson B-factor	36.46
R-merge	0.189 (1.180)
R-meas	0.196 (1.220)
R-pim	0.053 (0.325)
CC1/2	0.997 (0.844)
CC *	0.999 (0.957)
**Refinement**	
Reflections used in refinement	31,223 (3060)
Reflections used for R-free	1555 (159)
R-work	0.195 (0.268)
R-free	0.242 (0.322)
CCwork	0.951 (0.914)
CCfree	0.929 (0.794)
Number of non-hydrogen atoms	5570
macromolecules	5422
ligands	30
solvent	118
Protein residues	733
RMS (bonds)	0.007
RMS (angles)	0.99
**Validation**	
Ramachandran favoured (%)	97.26
Ramachandran allowed (%)	2.74
Ramachandran outliers (%)	0.00
Rotamer outliers (%)	1.37
Clash score	3.82
Average B-factor	43.73
macromolecules	43.76
Ligands	48.06
Solvent	41.42

* Statistics for the highest-resolution shell are shown in parentheses.

## Data Availability

The dataset presented in this study is available in this article. The crystal structure of LTL1 was deposited online at Protein Data Bank Japan (ID: 7V6D).

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
