# Peer review of "A Novel Lipase from Lasiodiplodia theobromae Efficiently Hydrolyses C8-C10 Methyl Esters for the Preparation of Medium-Chain Triglycerides’ Precursors"

_ijms, 2021, doi:10.3390/ijms221910339_

Round 1

Reviewer 1 Report

The manuscript describes a new lipase from L. theobromae discovered through zymographic-LC-MS analysis. The enzyme seems to prefer as substrates mono-glycerides and simple methyl esters over di- and triglycerides; moreover, derivatives of medium-chain fatty acids (C8-C10) are better accepted than long-chain fatty acids. Thus, Authors suggest that a possible application would be in the hydrolysis of caprylic and capric methyl esters affording the corresponding fatty acids, to be used in the manufacturing of medium-chain triacylglycerols of applicative interest.

In this respect, sometimes in the manuscript lipase LTL1 is mentioned as a possible tool for the synthesis of medium-chain triacylglycerols, while it actually would simply give access to the raw material for such an application, namely the free fatty acids (see for example lines 187-188). pH and temperature stability of the enzyme were also assessed. In order to gain insights into its substrate specificity, the 3D structure of the enzyme was solved by x-ray crystallography.

Since Authors point out the preference of the enzyme for monoglycerides and methyl esters, the term “monoester” is used throughout the manuscript. However, in such cases doubt often arises whether Authors refer to monoglycerides or to the simple methyl esters.

Error bars should be added in graphs of Fig. 2A; 3A, B and C; 4 and 5. Standard deviation values are lacking in table 1.

Figure 1C should be better explained: what do letters in the small boxes represent? How is it possible to align a peptide sequence, made up of amino acids, with a genome, made by base pairs?

Resolution of Fig. 6 appears to be rather poor. Fig. 6B and 6C appears quite confused and are hard to understand. Maybe the use of less colors can be advantageous in distinguishing which parts belong to which molecule. What do parts colored orange and violet refer to? They are neglected in the figure caption. Similar considerations apply for Fig. 6C.

Some minor points:

line 50: tricaprylin

line 54: a mixture containing

line 71: carbon and energy

line 167: higher than those of the commercial lipases

line 385: double dot at the end

line 399: towards the production

Reviewer 2 Report

Major points ;

1) Authors discovered a new lipase (LTL1) from a fungus which hydrolyses triglycerides with medium chain fatty acids (MCTs). They focused on MCTs production because MCTs are used for the treatment of AD patients. How will authors utilize LTL1 for MCTs production? Reverse reaction of LTL1? The important point is unclear.

2) Author identified the sequence of LTL1 gene. Compare the data with known lipase genes to make the difference clear (for example, by making  a phylogenetic tree). This makes LTL1 novelty realistic.

3) Figure 6 is not effective to understand the substrate specificity. At least, a substrate should be added to the protein structures.

Minor points;

4) In Figure 1A, it is difficult to observe the halo lipolytic zone.

5) Line 95; Authors identified 4 lipolitic bands on native PAGE and discovered LTL1 from the band with a lowest MW (~25kD). What happened to the other three bands? also obtained the same LTL1 ?

6) What is the best substrate of LTL1? Vinyl laurate ? not TG, it often happened to lipases. Explain these points.
